# Appendiceal Collision Tumors: An Institutional Case Series and Systematic Review of the Histologic Spectrum, Clinical Outcomes, and Management Strategies

**DOI:** 10.3390/diagnostics16010114

**Published:** 2025-12-30

**Authors:** Gizem Issin, Fatih Demir, Diren Vuslat Cagatay, Irem Guvendir, Hasan Aktug Simsek, Itir Ebru Zemheri

**Affiliations:** 1Department of Pathology, Erzincan Binali Yildirim University, Mengucek Gazi Training and Research Hospital, Erzincan 24100, Turkey; direncagatay@gmail.com; 2Department of Pathology, Faculty of Medicine, Canakkale Onsekiz Mart University, Canakkale 17020, Turkey; 3Department of Pathology, Faculty of Medicine, Duzce University, Duzce 81620, Turkey; fatihfnd@gmail.com; 4Department of Pathology, Health Science University Umraniye Training and Research Hospital, Istanbul 34764, Turkey; iremguvendir@hotmail.com (I.G.); ebruzemheri@gmail.com (I.E.Z.); 5Department of Pathology, Health Science University, Lutfu Kirdar Training and Research Hospital, Istanbul 34865, Turkey; 6Department of Pathology, Eskisehir City Hospital, Eskisehir 26080, Turkey; aktugsimsek@gmail.com; 7Department of Pathology, Istinye University, Istanbul 34396, Turkey

**Keywords:** appendiceal collision tumors, low-grade appendiceal mucinous neoplasm, neuroendocrine neoplasm, adenocarcinoma, goblet cell morphology, clinicopathologic features, systematic review

## Abstract

**Background/Objectives**: Appendiceal collision tumors (ACTs), defined by the coexistence of two or more histologically distinct neoplastic components within the appendix, are rare entities. We aimed to characterize their clinicopathologic features, management strategies, and outcomes by integrating an institutional case series with a systematic review of the literature. **Methods:** We retrospectively identified ACTs diagnosed at our institution and performed a PRISMA 2020-guided search of PubMed, Scopus, and Web of Science databases through May 2025 for case reports and case series. Two reviewers screened studies and extracted data on presentation, histologic composition, treatment, approaches and outcomes. **Results:** ACTs accounted for 4% of appendiceal tumors in our institution, all combining a neuroendocrine neoplasm (NEN) with a low-grade appendiceal mucinous neoplasm. The literature search identified 69 ACTs from 33 studies; pooled with our cases, 74 patients were evaluated. The most common pairings were NEN–appendiceal mucinous neoplasm (53%) and NEN–adenocarcinoma (26%), while three-component tumors were rare (*n* = 2). Early-stage tumors (pTis–pT1) were uniformly managed with appendectomy or limited resection, in line with established stage-based management algorithms for appendiceal neoplasms. Advanced-stage tumors (pT3–pT4) were treated according to the biologically dominant component, frequently with colectomy and, in high-risk mucinous disease, cytoreductive approaches. Across stages, outcomes appeared to be driven by the non-neuroendocrine component; a coexisting low-grade NEN did not independently confer worse prognosis. In ACTs with an adenocarcinoma component, goblet cell morphology was common, and outcomes appeared similar to those reported for non-collision appendiceal adenocarcinoma. **Conclusions:** ACTs represent a heterogeneous group in which prognosis is dictated by the non-neuroendocrine component and tumor stage. Low-grade NEN components appear biologically indolent, whereas adenocarcinoma and high-risk mucinous components have been observed to exhibit behavior similar to their solitary counterparts.

## 1. Introduction

Appendiceal neoplasms are uncommon, occurring in approximately 0.2–3.09% of all appendectomy specimens [1,2]. Neuroendocrine neoplasms (NENs) represent the most frequently encountered subtype, followed by low- and high-grade appendiceal mucinous neoplasms (LAMNs and HAMNs) and adenocarcinomas (ACs), which arise from distinct cellular lineages [3,4,5].

While most appendiceal tumors consist of a single histologic type, the simultaneous presence of two morphologically and immunophenotypically distinct neoplasms within the same appendix is exceptionally rare. Such lesions are referred to as appendiceal collision tumors (ACTs) [6,7]. In contrast to composite neoplasms, ACTs are characterized by the absence of histologic intermixing between their individual components.

Despite a clear morphologic definition, the etiopathogenesis of ACTs has not yet been fully elucidated. Several hypotheses have been proposed to explain their development. One suggests that a shared carcinogenic exposure or a permissive mucosal microenvironment may trigger independent neoplastic transformation in adjacent yet distinct cell lineages, consistent with the concept of field cancerization [8]. Another hypothesis proposes clonal divergence from a single multipotent progenitor cell, which may differentiate into neuroendocrine and epithelial phenotypes, potentially through an amphicrine intermediate [9]. In addition, the absence of histologic intermingling between tumor components has led to the interpretation of ACTs as coincidental collisions of two unrelated primary tumors arising independently within the confined anatomic space of the appendix [7]. In reported cases, ACTs most commonly present as recurring pairwise combinations. The most frequently reported combinations in the appendix involve NEN coexisting with mucinous neoplasms, mucinous neoplasms with ACs, or NEN occurring alongside ACs [6,7,10].

Owing to their rarity, ACTs have been predominantly described in isolated case reports or small case series, resulting in a limited understanding of their clinical behavior, prognostic determinants, and optimal management strategies. It remains unclear whether clinical outcomes are driven by the dominant tumor component or by the most aggressive histologic subtype.

In appendiceal neoplasms, management is largely determined by tumor type, stage, and the presence of peritoneal dissemination [11,12]. Simple appendectomy is generally sufficient for early-stage disease, whereas advanced tumors—particularly mucinous lesions—often require surgical escalation, including right hemicolectomy or cytoreductive surgery with hyperthermic intraperitoneal chemotherapy [11,12]. However, ACT-specific management recommendations remain ill-defined due to the limited and heterogeneous nature of the available evidence.

In this study, we aimed to characterize the clinicopathologic features of ACTs by combining a retrospective analysis of appendectomy specimens from our institution with a systematic review of the literature. By integrating original cases with previously published data, we sought to better delineate their histologic patterns, incidence, and clinical behavior.

## 2. Materials and Methods

### 2.1. Study Design and Population

This retrospective, two-center cohort study included all appendectomy specimens evaluated at Umraniye Training and Research Hospital and Erzincan Binali Yıldırım University Mengücek Gazi Training and Research Hospital between 1 January 2014 and 31 December 2022. All pathology reports issued during this period were screened sequentially.

Only primary appendiceal epithelial neoplasms, including NENs, ACs, LAMNs, and HAMNs, classified according to the 2019 WHO criteria, were eligible for inclusion. Exclusion criteria comprised: (i) non-epithelial lesions, including mesenchymal tumors (lipoma, leiomyoma, hemangioma, and neuroma) and hematolymphoid malignancies; (ii) secondary involvement of the appendix by metastases from extra-appendiceal primary tumors; (iii) cases lacking verifiable clinical or registry-based follow-up data to ascertain postoperative survival status; and (iv) cases with incomplete operative records. All eligible cases were retained for pathological reassessment and final analysis.

### 2.2. Pathologic Assessment and Data Collection

All eligible cases underwent detailed histopathologic reassessment. All available slides were independently reviewed by two experienced gastrointestinal pathologists, who were blinded to clinical information. Any discrepancies in component identification or staging were resolved by joint consensus. Tumors were classified according to the 2019 WHO criteria, and tumor size, anatomic location, depth of invasion, and pathologic stage were recorded following the College of American Pathologists (CAP) protocol and the 8th edition AJCC staging system [13,14]. Lesions previously reported as “goblet cell carcinoid” or “goblet cell neuroendocrine carcinoma” were reclassified as goblet cell adenocarcinoma in accordance with current WHO terminology. Clinical data, including patient demographics, presenting symptoms, operative procedure, use of adjuvant therapy, recurrence status when available, and follow-up duration, were retrieved from electronic medical records using a standardized data capture form. Follow-up information was obtained from institutional clinical records and supplemented, when necessary, by registry-based verification of postoperative vital status through national death notification systems. Overall survival was defined as the interval from the date of surgery to death or last confirmation of survival.

### 2.3. Systematic Literature Search and Study Selection

A systematic search of PubMed/MEDLINE, Scopus, Web of Science, and Google Scholar was performed from database inception through 1 May 2025. The search strategy combined the terms “appendiceal collision tumor”, “collision tumor appendix”, “synchronous appendiceal tumors”, “double tumor appendix”, and “triple tumor appendix”, without language or publication date restrictions. All records were imported into EndNote X9 and de-duplicated.

Two reviewers (I.G. and D.F.) independently screened titles and abstracts, followed by full-text assessment against prespecified eligibility criteria. Any discrepancies were resolved by consensus. All case reports and case series describing appendiceal collision tumors were eligible for inclusion, irrespective of the extent of clinical or pathological detail provided. Reference lists of included articles were screened to identify additional cases.

Data extraction was performed independently by two reviewers using a standardized data collection template. For each case, the following variables were recorded when available: age, sex, histologic components, tumor size, surgical procedure, use of adjuvant therapy, follow-up duration, and outcome, when available. Data were recorded as reported; missing or unclear items were not imputed.

Given that the available evidence consisted predominantly of single-case reports and small case series, no formal risk-of-bias or certainty assessment was undertaken. A descriptive synthesis was therefore performed to summarize clinicopathologic characteristics and treatment patterns. The study selection process is presented in the PRISMA 2020 flow diagram (Figure 1).

### 2.4. Statistical Analysis

Statistical analyses were performed using SPSS version 18.0 (IBM Corp., Armonk, NY, USA). Continuous variables are summarized as mean ± standard deviation, and categorical variables as counts and percentages. Normality was assessed using the Shapiro–Wilk test. Comparisons of continuous variables were conducted using the Mann–Whitney U test for two-group comparisons (ACTs vs. non-collision appendiceal neoplasms) and the Kruskal–Wallis test for analyses involving more than two histologic subgroups, with Dunn–Bonferroni adjustment applied for post hoc pairwise comparisons. Categorical variables were analyzed using Fisher’s exact test.

Overall survival was high across the cohort, with very few observed events. Given the low number of outcome events, formal survival analyses such as Kaplan–Meier estimation were not statistically informative and were therefore not performed. Survival outcomes are reported as follow-up duration and vital status at last ascertainment. All statistical tests were two-sided, and a *p* value < 0.05 was considered statistically significant.

### 2.5. Ethical Approval

This study was conducted in accordance with the ethical standards of the institutional and national research committee and the 1964 Helsinki Declaration and its later amendments or comparable ethical standards. Ethical approval was obtained from the Erzincan Binali Yildirim University Ethics Committee (Approval No.: 19/1).

## 3. Results

### 3.1. Institutional Cohort

In our cohort, appendiceal tumors were identified in 126 of 9252 appendectomy specimens, corresponding to a frequency of 1.36%. Of these, 121 cases (96%) had a single tumor type, with NENs being the most common (61 cases, 48%), followed by appendiceal mucinous neoplasms (AMNs; LAMN or HAMN) (54 cases, 43%) and AC (6 cases, 5%). ACTs were detected in 5 cases (4% of appendiceal tumors), representing 0.05% of all appendectomy specimens. Detailed clinicopathologic characteristics and clinical outcomes of ACT cases are summarized in Table 1, and representative histologic images of the main tumor components are shown in Figure 2.

This figure illustrates the histopathological diversity and proportional distribution of appendiceal tumors in our cohort (*n* = 126). Representative hematoxylin and eosin-stained micrographs of the major tumor categories are shown. Low-grade appendiceal mucinous neoplasm (LAMN), neuroendocrine neoplasm (NEN), adenocarcinoma (AC), and appendiceal collision tumors (ACTs), including coexisting NEN and LAMN. The pie chart shows the overall frequency of each tumor type: NEN (48%), LAMN (43%), AC (5%), and ACTs (4%).

#### Comparative Analysis of Appendiceal Collision Tumors and Non-Collision Appendiceal Neoplasms in the Institutional Cohort

Among ACT cases, most patients were female (3/2; F/M ratio: 1.5), whereas sex distribution among non-collision appendiceal neoplasms was nearly balanced (56 females, 65 males; F/M ratio: 0.86). Within non-collision cases, LAMN was slightly more common in males (29 vs. 25; F/M ratio: 1.16), while NEN showed a nearly equal distribution (32 males vs. 29 females; F/M ratio: 1.10). Overall, ACT cases demonstrated a female predominance; however, the sex distribution did not differ significantly between ACTs and non-collision appendiceal neoplasms (Fisher’s exact test, *p* = 1.000), likely reflecting the small number of ACTs.

The mean age was comparable between ACT patients (40.2 years, range: 23–60) and those with non-collision appendiceal neoplasms (40.04 years, range: 10–83). When stratified by histologic subtype, LAMN/HAMN cases showed the highest mean age (49.74 years), whereas patients with NEN were markedly younger (mean 26.64 years). Patients with ACTs, characterized by coexisting NEN and LAMN components, were diagnosed at a significantly older age than patients with NEN alone (*p* < 0.001). Detailed age, sex, and stage distributions for NEN, LAMN, AC, and ACT groups are provided in Table 2.

The median follow-up duration for the entire cohort was 68 months (range: 5–158), and 80.2 months (range: 20–128) for ACT cases. In the STT group, 29 patients underwent right hemicolectomy, and chemotherapy was administered to those with AC. Additional operations were performed for synchronous malignancies, including rectal adenocarcinoma (*n* = 4), sigmoid colon adenocarcinoma (*n* = 1), descending colon adenocarcinoma (*n* = 1), and ovarian carcinoma (*n* = 1). During follow-up, four deaths occurred in the non-collision group.

All ACTs were incidental findings in appendectomy specimens performed for suspected acute appendicitis. Tumor staging in ACT cases was as follows: pT1/pTis (two cases, corresponding to NEN and LAMN, respectively), pT1/pT4 (one case), pT2/pT3 (one case), and pT3/pTis (one case). One patient with ACT underwent right hemicolectomy due to a NEN component measuring >2 cm, and lymph node metastasis was identified in the resection specimen. No additional treatment was administered to the remaining ACT patients. During follow-up, no recurrences or deaths were observed in the ACT group.

### 3.2. Literature Synthesis

#### 3.2.1. Demographics and Clinical Presentation

A total of 69 ACT cases were identified in the literature. A summary of the main clinicopathologic features is shown in Table 3, with additional details available in Appendix A. Among these patients, 38 were female and 31 were male, giving a female-to-male ratio of 1.2. The mean age at diagnosis was 52.8 years (range, 18–82), and the median age was 54 years.

Clinical presentation was described in 52 of the 69 cases (75.4%). The most frequently reported symptoms were nonspecific abdominal pain or discomfort (12 cases, 23.1%) and localized right lower quadrant pain (12 cases, 23.1%). Ten patients (19.2%) presented with a clinical picture suggestive of acute appendicitis, and incidental diagnoses during unrelated procedures were documented in 8 cases (15.4%). Palpable abdominal or pelvic masses prompted evaluation in 6 patients (11.5%), while three cases (5.8%) were investigated for alternative indications, including small bowel obstruction, mucin accumulation within a hernia sac, or elevated carcinoembryonic antigen levels. Clinical information was not available for the remaining 17 cases (24.6%).

#### 3.2.2. Histological Composition

Information on histologic composition was available for 56 patients. The most frequent combination was NEN coexisting with LAMN or HAMN, observed in 41 cases. This was followed by NEN combined with AC in 13 cases and AC combined with LAMN or HAMN in 10 cases. In two rare cases, all three tumor components—NEN, LAMN/HAMN, and AC—were identified within the same appendix.

#### 3.2.3. Grading and Staging

Among NEN components (*n* = 44), most were low-grade (G1; 37/44, 84%), while six cases (13.6%) were intermediate-grade (G2) and one case (2.3%) fulfilled criteria for neuroendocrine carcinoma (NEC). Staging information was available for 37 NENs: 14 tumors (37.8%) were pT1, 5 (13.5%) were pT2, 14 (37.8%) were pT3, and 4 (10.8%) were pT4.

In cases with an AC component, histologic subtype was reported in 23 patients. Fifteen were classified as goblet cell adenocarcinoma, 4 as mucinous adenocarcinoma, and 4 as conventional colonic-type adenocarcinoma. Staging data were available in 11 of these cases: one tumor was pTis, one was pT2, five were pT3, and four were pT4.

Among mucinous neoplasms, 38 patients were diagnosed with LAMN and 3 with HAMN. These tumors most often presented at early stages. Staging information was available for 38 cases: 23 (60.5%) were pTis, 4 (10.5%) were pT3, and 11 (28.9%) were pT4.

Taken together, these findings indicate that most NEN components are low grade, AC components frequently show goblet cell morphology and advanced stage, and mucinous neoplasms tend to present in situ, with a subset progressing to higher pT categories.

#### 3.2.4. Treatment and Outcome

##### NEN and LAMN Combination

In cases with combined NEN and LAMN components, treatment generally followed the pathologic stage of each component. Among early-stage tumors, particularly those with pT1 NEN and pTis LAMN, appendectomy was the most common procedure. Seven patients with this combination were treated with appendectomy alone, without adjuvant therapy, and none of them experienced recurrence or disease-related death during follow-up. One additional patient with pT1 NEN and pTis LAMN underwent right hemicolectomy without HIPEC and likewise had an uneventful course.

For more advanced combinations, one patient with pT1 NEN and pT3 LAMN was managed with RHC and remained recurrence-free. In another case with the same component stages, RHC was combined with HIPEC, also without subsequent recurrence or disease-related mortality. A further patient with pT1 NEN and pT4 LAMN was treated with appendectomy alone and showed no evidence of disease during follow-up. However, in two additional cases with pT4 LAMN components, right hemicolectomy and HIPEC were performed and both patients died during follow-up.

##### AC and LAMN Combination

Among reported cases with combined AC and LAMN components, tumor stage and treatment were highly variable. LAMN components were staged as pTis in three cases and as pT4 in two cases. When staging was specified, AC components were most often classified as pT3 or pT4.

Treatment modalities included right hemicolectomy in two patients, one of whom also received CRS and HIPEC. One patient underwent ileocecal resection with regional lymph node dissection, and two patients were managed with appendectomy alone. In four cases, the type of surgical procedure was not reported.

Follow-up information was limited. One patient treated with right hemicolectomy, CRS, and HIPEC was alive and disease-free at six months, whereas another patient who underwent CRS and chemotherapy died of disease within seven months. For the remaining cases, follow-up and survival outcomes were not described.

##### AC and NEN Combination

Treatment approaches included right hemicolectomy in eight cases, two of which were combined with pelvic resection. Two patients underwent more limited procedures: one had appendectomy alone, and another underwent appendectomy with cecal resection. Chemotherapy alone was administered in one patient.

Follow-up data were available for ten cases. Two patients died of disease after surgery, at 6 and 14 months, respectively, while seven patients were reported to be alive and disease-free at follow-up intervals ranging from 6 to 65 months. In the remaining cases, follow-up duration or survival status was not specified.

## 4. Discussion

ACTs are rare entities characterized by the coexistence of two or more histologically distinct neoplastic components. Their biological behavior and optimal management remain poorly defined, largely because of the limited number of reported cases and the lack of standardized diagnostic or therapeutic frameworks. In our institutional cohort, ACTs accounted for 4% of all appendiceal tumors, and all cases consisted of combined LAMN and NEN components. Beyond our institutional series, 69 published ACT cases were identified, resulting in a pooled total of 74 cases with available clinical, histologic, and outcome data.

Across this expanded cohort, the most frequent tumor combination was NEN with LAMN or HAMN, accounting for 53% of cases. Tumors that included an AC component, either with NEN or with LAMN/HAMN, comprised 43% of cases. A small minority of tumors (*n* = 2) contained all three histologic components (NEN, LAMN/HAMN, and AC).

The mean age at diagnosis in our institutional cohort was 40.2 years, which is lower than the mean age of 52.8 years reported across published cases. This difference likely reflects the predominance of NEN and LAMN combinations in our series, whereas AC-containing tumors, which are more frequent in the literature, tend to present at older ages. The female-to-male ratio in our cohort was 1.5, compared with 1.2 in the pooled literature cases [16,17,18,19,20,21,22,23,24,25,26,27,28,29,30,31,32,33,34,35,36,37,38,39,40,41,42,43,44].

Collectively, ACT cases from our cohort and the literature show clinical presentations that largely overlap with those reported for non-collision appendiceal neoplasms [2]. Most patients presented with symptoms compatible with acute appendicitis or with nonspecific abdominal pain. These findings are consistent with the well-described clinical behavior of appendiceal mucinous neoplasms and NENs [1,2]. Within this context, ACTs do not appear to display distinctive clinical features that would permit reliable preoperative suspicion or allow meaningful radiologic or symptom-based differentiation.

In terms of histologic composition, ACTs were most frequently composed of a combination of NEN and LAMN, followed by NEN with AC and AC with LAMN. NENs and LAMNs are also common among non-collision appendiceal neoplasms, whereas ACs are comparatively rare [1,2]. In that setting, the most common AC subtypes are mucinous and conventional intestinal-type adenocarcinomas, whereas goblet cell adenocarcinomas are uncommon [45,46]. By contrast, in ACTs, AC components most often display goblet cell morphology.

The frequent predominance of goblet cell-type morphology among adenocarcinoma components in ACTs warrants further consideration. Several, not mutually exclusive, factors may contribute to this observation. From a biological perspective, the appendix differs from the colorectum in its mucosal composition, with a relatively prominent goblet cell population, a higher density of neuroendocrine cells, and a lymphoid tissue-rich microenvironment [47,48]. Such features may support amphicrine or dual-lineage differentiation. In keeping with this concept, goblet cell-type neoplasms are intrinsically more common in the appendix than in the colorectum and occupy a phenotypic interface between glandular and neuroendocrine differentiation, frequently showing overlapping morphologic and immunophenotypic features [49,50]. In addition, diagnostic and classification-related factors may play a role, as morphologically distinctive combinations involving neuroendocrine and goblet cell-rich components are more readily recognized and reported. Moreover, the recent reclassification of goblet cell carcinoid within the adenocarcinoma spectrum may have further inflated their apparent representation among reported ACTs [13]. Taken together, these considerations suggest that the prominence of goblet cell morphology in ACTs likely reflects a combination of biological predisposition and diagnostic or classification-related effects rather than a single underlying mechanism.

Although molecular data specific to ACTs remain limited, insights from the broader appendiceal tumor literature provide a useful biological framework for interpreting the frequent goblet cell–type morphology observed in these lesions. Low-grade appendiceal mucinous neoplasms are predominantly driven by KRAS- and GNAS-associated mucinous programs, whereas progression toward invasive adenocarcinoma more commonly involves additional alterations affecting tumor suppressor and differentiation pathways, including TP53 and TGF-β/SMAD signaling [47,51]. Goblet cell-type adenocarcinomas appear to occupy an intermediate biological position, showing reduced dependence on canonical mucinous drivers alongside partial enrichment of pathways associated with epithelial progression and lineage plasticity, including WNT and NOTCH signaling [49,50,51]. In contrast, appendiceal NENs generally follow distinct molecular trajectories, most often involving pathways related to neuroendocrine differentiation and cell-cycle regulation rather than mucinous driver programs [52]. Accordingly, the coexistence of epithelial and neuroendocrine components in ACTs is more plausibly explained by overlapping differentiation programs and a shared permissive mucosal microenvironment than by identical oncogenic driver events [13,51].

Current clinical evidence indicates that tumor stage and the most aggressive histologic component remain the primary determinants of prognosis and management in ACTs. In appendiceal neoplasms, treatment strategies and prognosis are primarily determined by tumor type and stage at diagnosis. Low-grade NENs, which are frequently identified incidentally, usually follow an indolent clinical course [53]. For tumors <2 cm and confined to the submucosa (typically pT1), appendectomy is generally sufficient, and long-term outcomes are excellent. In contrast, tumors ≥2 cm carry a higher risk of lymph node metastasis, reported in up to 64% of cases, and may warrant right hemicolectomy depending on additional factors such as lymphovascular invasion, mesoappendiceal extension, or a high proliferative index [54,55,56]. However, recent studies have questioned the survival benefit of right hemicolectomy in larger tumors, with some series showing no significant improvement in disease-specific survival even for NENs > 2 cm [57,58].

In ACTs that include a NEN component, clinical management generally parallels established treatment approaches for non-collision appendiceal NENs [53]. In both our cohort and the reviewed literature, most NEN components were low grade (G1), and pT1 and pT3 were the most frequent stages. Accordingly, limited procedures such as appendectomy were often sufficient, and outcomes were consistent with those reported for appendiceal NENs in general. Our findings further suggest that when the accompanying LAMN or AC component is confined to an early pathologic stage (e.g., pTis or pT1), overall clinical behavior remains comparable to that expected for appendiceal neoplasms of a single histologic type. In both our institutional series and the pooled literature, these early-stage secondary components did not appear to substantially modify the disease course. Within this context, and based on currently available data, there is no clear evidence that low-stage ACTs require routine escalation of surgical management solely because of the presence of a second histologic component.

When evaluating cases with a NEN component accompanied by advanced-stage AMNs, a total of 13 collision tumors were identified, including four pT3 and nine pT4 tumors. Six patients underwent CRS with HIPEC, whereas the remaining seven were treated with appendectomy or right hemicolectomy alone and remained progression-free during follow-up (median follow-up, 11 months; range, 6–128 months). Notably, two patients in the CRS with HIPEC group ultimately died of disease. Despite the limited sample size and incomplete staging information, the treatment approaches observed in these cases appeared to follow established management protocols for AMNs, without obvious modification based on the presence of a NEN component [58,59]. Reported 5-year overall survival rates for disseminated low-grade AMNs with uniform histology range between 60% and 70%, which is comparable to the 66% survival rate observed in our NEN plus AMN subgroup treated with CRS and HIPEC [60]. Taken together, these observations indicate that the coexistence of a NEN component does not seem to confer additional adverse prognostic impact beyond that of the mucinous neoplasm itself, although this conclusion is limited by the small number of available cases.

When advanced-stage appendiceal AC (pT3–pT4) containing a NEN component were analyzed (*n* = 7), crude mortality was 42.9%, which falls within the broad mortality rates reported for advanced-stage non-collision appendiceal ACs treated with comparable protocols [61,62]. Treatment patterns in this subgroup, including the use of CRS with HIPEC in 44% of patients, were comparable to those described for advanced stage appendiceal adenocarcinoma, suggesting that the presence of a NEN component neither systematically prompted more aggressive management nor clearly worsened prognosis. Nevertheless, these findings should be interpreted with caution given the small number of cases and incomplete staging and follow-up data. Owing to the paucity of staging and outcome information, no meaningful conclusions could be drawn for ACTs composed of AC and LAMN components.

Based on the collective evidence in this study, the presence of two histologic components, particularly combinations involving NENs and mucinous neoplasms, does not appear to substantially alter clinical behavior, treatment decisions, or oncologic outcomes compared with solitary tumor types of similar histology and stage. In both early- and advanced-stage settings, management strategies largely mirrored those employed for single-component tumors. Importantly, no consistent signal emerged to suggest that the addition of a NEN component confers a significant adverse prognostic effect, even in cases that include advanced AC or AMNs.

This study should be interpreted with consideration of certain limitations, mainly related to its retrospective design and to the rarity of ACTs. The number of cases within each histologic subgroup was small, and staging information was incomplete in a substantial proportion of reported tumors. Follow-up durations varied across patients, and the lack of longitudinal outcome data for certain component combinations, such as AC–LAMN tumors, limited our ability to fully assess their clinical significance.

Despite these constraints, the integration of institutional data with a systematic review of the literature allowed us to assemble one of the largest clinically annotated ACT datasets reported to date. Although definitive recommendations regarding optimal management cannot yet be established, our findings provide a pragmatic overview of real-world treatment patterns and support a component- and stage-based approach to interpretation. They also underscore the need for standardized pathologic reporting and prospective multicenter collaboration, including molecular and outcome-focused studies, to better define the biological behavior and therapeutic implications of ACTs.

## 5. Conclusions

ACTs are rare neoplasms for which evidence-based guidance remains limited. In this combined institutional and literature-based analysis, ACTs generally appeared to mirror the clinical behavior and management patterns of non-collision appendiceal neoplasms of comparable histology and stage. In early-stage disease, outcomes were favorable with limited surgical approaches, whereas in advanced-stage settings the presence of a NEN component did not clearly worsen prognosis. Overall, our findings support the concept that management decisions in ACTs should be guided primarily by the most aggressive histologic component, in conjunction with tumor stage and other established prognostic factors, rather than by the collision architecture itself.

## Figures and Tables

**Figure 1 diagnostics-16-00114-f001:**
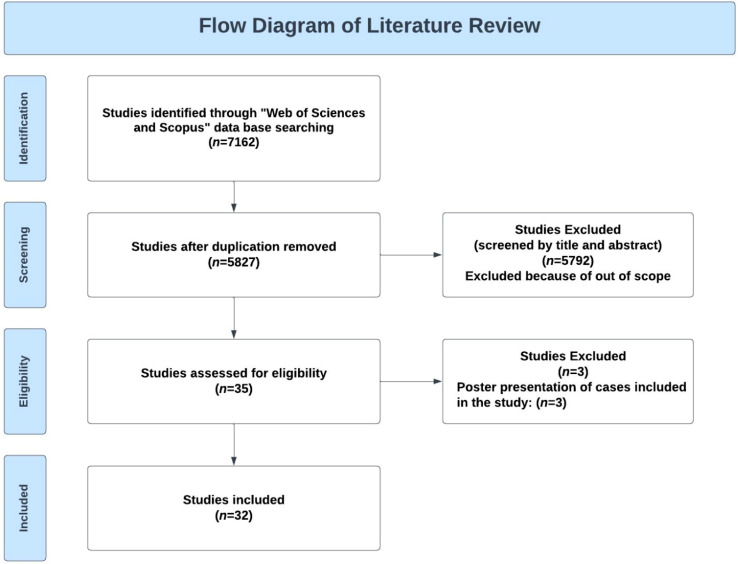
Flow diagram of the systematic literature review and article selection process.

**Figure 2 diagnostics-16-00114-f002:**
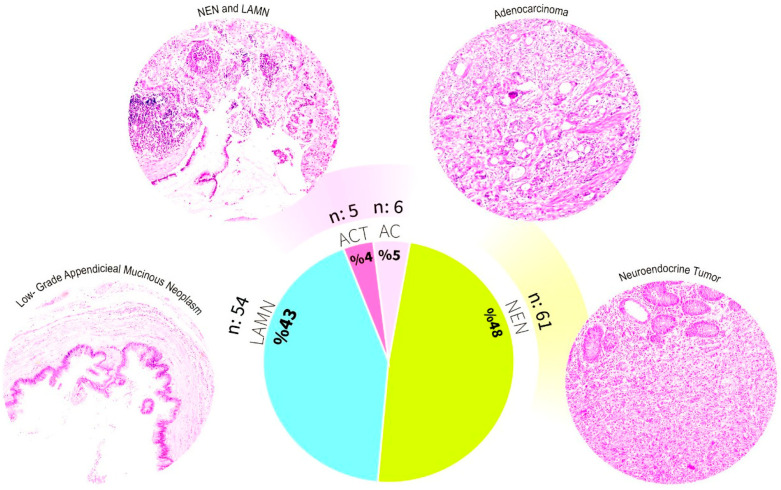
Histological classification and distribution of appendiceal tumors in our cohort.

**Table 1 diagnostics-16-00114-t001:** Clinicopathologic characteristics, component-specific staging, treatment, and follow-up of institutional ACT cases.

No	Age	Sex	Presentation	NEN (Grade / pT)	LAMN (pT)	Treatment	Follow-Up (Months)
1	56	Female	RLQ pain	G 1 / pT1	pTis	Simple appendectomy	Alive (103)
2	23	Male	RLQ pain	G 1 / pT1	pTis	Simple appendectomy	Alive (124)
3	31	Female	RLQ pain	G 1 / pT1	pT3	Simple appendectomy	Alive (128)
4	31	Male	RLQ pain	G 1 / pT2	pT4	Right hemicolectomy	Alive (20)
5	60	Female	RLQ pain	G 1 / pT3	pTis	Simple appendectomy	Alive (26)

NEN, neuroendocrine neoplasm; LAMN, low-grade appendiceal mucinous neoplasm; RLQ, right lower quadrant; G, Grade; pT, pathological tumor stage.

**Table 2 diagnostics-16-00114-t002:** Clinicopathologic Features of Appendiceal Tumors by Histologic Subtype in the Institutional Cohort.

Variable	NEN	LAMN/HAMN	AC	ACT		Total	*p*
Number of cases, *n*	61	54	6	5		126	
Sex							1.00 *
Female, *n* (%)	29 (47.5%)	25 (46.3%)	2 (33.3%)	3 (60.0%)		59 (46.8%)	
Male, *n* (%)	32 (52.5%)	29 (53.7%)	4 (66.7%)	2 (40.0%)		67 (53.2%)	
Mean age (years)	26.69	49.74	58.00	40.20		38.60	* p * < 0.001 **
Status							
Alive, *n*	61 (100%)	52 (96.3%)	4 (66.7%)	5 (100%)		122 (96.8%)	
Dead, *n*	0 (0%)	2 (3.7%)	2 (33.3%)	0 (0%)		4 (3.2%)	
	NEN	LAMN/HAMN	AC	ACT–NEN component	ACT–LAMN component		
Tumor stage distribution							
pTis	–	25	–	–	3		
pT1	36	–	2	3	–		
pT2	4	–	1	1	–		
pT3	16	18	3	1	1		
pT4	5	11	–	–	1		

Abbreviations: NEN, neuroendocrine neoplasm; LAMN, low-grade appendiceal mucinous neoplasm; HAMN, high-grade appendiceal mucinous neoplasm; AC, adenocarcinoma; ACTs, appendiceal collision tumors. * Fisher’s exact test; ** Kruskal–Wallis test.

**Table 3 diagnostics-16-00114-t003:** Summary of Appendiceal Collision Tumor Cases Identified in the Literature Review.

Study	*n*	Tumor Components (pT Stage)	Surgical Approach	Outcome
Durowoju et al. 2025 [15]	15	NEN/LAMN/HAMN/AC (various pT)	Appendectomy/RHC	DCD 1/15; Alive 14/15
Hirose et al. 2025 [16]	1	LAMN pTis/Goblet cell AC pT3	Ileocecal resection	Alive
H. AlAwfi et al. 2024 [17]	1	NEN pT1/LAMN pTis	RHC	Alive
Viel et al. 2023 [7]	2	NEN pT1–2/LAMN pTis	Appendectomy	Alive
Morillo et al. 2023 [18]	1	NEN pT3/HAMN pTis	Appendectomy	Alive
Gupta et al. 2023 [19]	1	NEN pT2/LAMN pT4	RHC + HIPEC	DCD
Rahman et al. 2022 [20]	1	NEN pT1/LAMN pT3/AC pT2	RHC	Alive
Syrine Moussa et al. 2022 [21]	1	NEN (pT NS)/LAMN pTis	Total colectomy	NS
Oka et al. 2022 [22]	1	AC pTis/HAMN pTis	Appendectomy	Alive
Melendez et al. 2022 [23]	2	Goblet cell AC pT3–4/LAMN pT4	RHC ± CRS/HIPEC	Alive 1/2; NS 1/2
Ekinci et al. 2021 [24]	1	NEN (pT NS)/LAMN (pT NS)	Appendectomy	Alive
Ruiz et al. 2021 [25]	1	Mucinous AC pT2/NEN pT3	Appendectomy + cecal resection	Alive
Villa et al. 2021 [10]	1	NEN pT3/LAMN pTis	RHC	Alive
Cafaro et al. 2020 [26]	1	NEN pT3/LAMN pTis	Appendectomy	Alive
Carboni et al. 2020 [27]	1	NEN pT3/LAMN pTis/Goblet cell AC (NS)	Appendectomy	Alive
Sholi et al. 2019 [28]	1	NEN pT4/LAMN pT4	RHC	Alive
Chinen et al. 2019 [29]	1	Goblet cell AC pT4/LAMN (pT NS)	CRS + CHT	DCD
Hajjar et al. 2019 [30]	1	NEN pT3/LAMN pT4, M1	RHC + CRS + HIPEC	Alive
Yeh et al. 2018 [31]	1	AC pTis/NEN (pT NS)	Appendectomy	NS
Sato et al. 2018 [32]	1	Goblet cell AC pT3/LAMN pTis	NS	NS
R. Das et al. 2017 [33]	1	NEN (pT NS)/LAMN pTis	RHC	NS
Sugarbaker et al. 2016 [34]	2	NEN pT1–2/LAMN pT3, M1	RHC ± CRS/HIPEC	Alive
Tan et al. 2015 [35]	1	NEN (pT NS)/LAMN pTis	Appendectomy	NS
Baena-del-Valle et al. 2015 [36]	2	NEN (pT NS)/LAMN pT4	Appendectomy + CRS/HIPEC	Alive 1/2; DCD 1/2
Ng et al. 2014 [37]	2	LAMN/Goblet cell AC (pT NS)	NS	NS
Dellaportas et al. 2014 [38]	1	NEN pT4/LAMN pTis	RHC	Alive
Singh et al. 2010 [39]	1	Mucinous AC pT3, M1/NEN pT3	CHT	DCD
Chetty et al. 2010 [40]	5	Goblet cell AC/NEN (various pT)	Appendectomy/RHC	Alive 3/5; DCD 1/5; NS 1/5
Alsaad et al. 2009 [41]	1	LAMN pTis/Goblet cell AC pT3	RHC	NS
Rossi et al. 2004 [42]	1	Mucinous AC pT3/NEC (pT NS)	RHC	Alive
Al-Talib et al. 1995 [43]	2	LAMN pT4/Goblet cell AC (pT NS)	NS	NS
Carr et al. 1995 [6]	13	NEN/Goblet cell AC/Mucinous neoplasms (various pT)	NS	NS
Sjövall et al. 1985 [44]	1	AC (pT NS)/NEN (pT NS)	RHC	Alive

Abbreviations: AC, adenocarcinoma; CRS, cytoreductive surgery; DCD, dead of disease; HIPEC, hyperthermic intraperitoneal chemotherapy; LAMN, low-grade appendiceal mucinous neoplasm; HAMN, high-grade appendiceal mucinous neoplasm; NEN, neuroendocrine neoplasm; NS, not specified; pT, pathological tumor stage; RHC, right hemicolectomy. pT stage is shown separately for each component. For heterogeneous case series, staging is summarized as “various pT” and detailed in Appendix A.

## Data Availability

The data that support the findings of this study are available from the corresponding author upon reasonable request.

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
