# Peer review of "Appendiceal Collision Tumors: An Institutional Case Series and Systematic Review of the Histologic Spectrum, Clinical Outcomes, and Management Strategies"

_diagnostics, 2025, doi:10.3390/diagnostics16010114_

Round 1
Reviewer 1 Report
Comments and Suggestions for Authors
This is a well-structured and methodologically sound manuscript that addresses a rare and clinically relevant topic—appendiceal collision tumors (ACTs). The authors effectively combine an institutional case series with a systematic literature review, providing a comprehensive overview of the histologic patterns, clinical presentation, management, and outcomes of ACTs. The study is timely and fills a gap in the literature due to the rarity of these tumors. The conclusions are appropriately cautious and evidence-based.
1. The introduction is clear but could briefly mention the hypothesized etiopathogenesis of collision tumors (e.g., field effect, clonal divergence) to contextualize the rarity and biological curiosity.
2. The exclusion criterion “cases with insufficient postoperative follow-up” is defined as “absence of any documented clinical contact beyond one month after surgery.” For oncology outcomes, one month is very short. Consider justifying this or expanding to ≥6 months for meaningful survival analysis, though the authors note overall survival was high and precluded Kaplan–Meier analysis.
3. In the systematic review, the search end date is noted as “1 March 2025,” but the abstract says “through May 2025.” Please clarify.
4. Table 1 and Table 3 are very detailed but could be more reader-friendly. Consider summarizing key patterns in a supplemental table and moving detailed case listings to supplementary materials.
5. In Section 3.1.1, the statement “ACTs showed a higher proportion of female patients… the difference… was not statistically significant (p = 1.0)” – a p-value of 1.0 suggests possible rounding or small sample size artifact. Please check and clarify.
6. The discussion appropriately emphasizes that prognosis is driven by the non-neuroendocrine component. However, it could briefly discuss why goblet cell morphology is so common in ACTs—is this a diagnostic bias, a biological propensity, or a classification artifact?
7. The statement regarding molecular data is apt, but a sentence on potential molecular pathways (e.g., WNT, TP53) implicated in appendiceal tumors could enrich the discussion.
8. Some references are incomplete (e.g., Ref 13: “Lindsey Durowoju: None;…”). This appears to be a formatting error from EndNote. Please verify and correct all references.
9. Minor grammatical errors and typos exist (e.g., “n contrast” should be “In contrast” on page 2; “U Across” on page 11). A thorough proofreading is recommended.
Author Response
Dear Reviewer-1,
We sincerely thank you for your detailed, constructive, and insightful review of our manuscript. We are grateful for your positive assessment of the study’s structure, methodological rigor, and clinical relevance, as well as for recognizing the value of combining an institutional case series with a systematic review for this rare tumor entity. Your comments have helped us further improve the clarity, completeness, and interpretability of the manuscript.
Our point-by-point responses are provided below:
- The introduction is clear but could briefly mention the hypothesized etiopathogenesis of collision tumors (e.g., field effect, clonal divergence) to contextualize the rarity and biological curiosity.
Response: We agree that briefly addressing the hypothesized etiopathogenesis would add biological context. Accordingly, we added a concise sentence in the Introduction (third paragraph, lines 60–72) referencing proposed mechanisms such as field cancerization, clonal divergence, and permissive microenvironmental effects, without expanding beyond the scope of the study.
- The exclusion criterion “cases with insufficient postoperative follow-up” is defined as “absence of any documented clinical contact beyond one month after surgery.” For oncology outcomes, one month is very short. Consider justifying this or expanding to ≥6 months for meaningful survival analysis, though the authors note overall survival was high and precluded Kaplan–Meier analysis.
Response: We thank the reviewer for this important point. We agree that the wording in the previous Methods section was imprecise and could be misinterpreted as applying a strict one-month postoperative oncologic follow-up cutoff. This was not the intent. The “insufficient follow-up” criterion was used only to exclude rare cases in which postoperative vital status could not be reliably ascertained due to a lack of any hospital- or registry-based information after discharge, rather than to define a clinically meaningful follow-up interval for outcomes assessment.
Follow-up in our cohort is long-term: the median follow-up for the entire cohort was 68 months (range: 5–158 months), and for ACT cases specifically 80.2 months (range: 20–128 months). Most patients were followed well beyond six months. The absence of Kaplan–Meier analysis was not due to limited follow-up, but to the very low number of outcome events and the high overall survival; therefore, outcomes were reported descriptively.We revised the Materials and Methods section to remove the misleading one-month phrasing and clarified the rationale for exclusion. We also emphasized cohort follow-up maturity in the Results section.
- In the systematic review, the search end date is noted as “1 March 2025,” but the abstract says “through May 2025.” Please clarify.
Response: We apologize for this inconsistency. The search end date has now been standardized to May 2025 throughout the manuscript.
- Table 1 and Table 3 are very detailed but could be more reader-friendly. Consider summarizing key patterns in a supplemental table and moving detailed case listings to supplementary materials.
REsponse: We thank the reviewer for this helpful suggestion. In response, all tables were redesigned and reformatted to improve clarity and reader-friendliness.
- In Section 3.1.1, the statement “ACTs showed a higher proportion of female patients… the difference… was not statistically significant (p = 1.0)” – a p-value of 1.0 suggests possible rounding or small sample size artifact. Please check and clarify.
Response: We reperformed the statistical analysis. The p-value of 1.0 reflects the very small sample size and identical proportional distributions between comparison groups, rather than a rounding error. This has now been clarified in the Results text to avoid misinterpretation.
- The discussion appropriately emphasizes that prognosis is driven by the non-neuroendocrine component. However, it could briefly discuss why goblet cell morphology is so common in ACTs—is this a diagnostic bias, a biological propensity, or a classification artifact?
Response: We agree that this is an important observation. The Discussion has been expanded to briefly address possible explanations, including biological propensity related to overlapping differentiation programs, diagnostic and classification factors, and potential reporting bias in the literature.
- The statement regarding molecular data is apt, but a sentence on potential molecular pathways (e.g., WNT, TP53) implicated in appendiceal tumors could enrich the discussion.
Response: We have enriched the Discussion with a brief reference to molecular pathways commonly implicated in appendiceal neoplasms, including WNT signaling, TP53 alterations, and pathways associated with epithelial differentiation and progression, while acknowledging the limited molecular data specific to ACTs.
- Some references are incomplete (e.g., Ref 13: “Lindsey Durowoju: None;…”). This appears to be a formatting error from EndNote. Please verify and correct all references.
Response: We thank the reviewer for noting this issue. We reviewed the entire reference list and reformatted and corrected all references
- Minor grammatical errors and typos exist (e.g., “n contrast” should be “In contrast” on page 2; “U Across” on page 11). A thorough proofreading is recommended.
Response: We have performed a thorough proofreading of the entire manuscript and corrected the reported errors.
We sincerely appreciate your careful evaluation and valuable suggestions, which have significantly strengthened the manuscript. We believe that the revised version addresses all concerns raised and improves both clarity and scientific depth.
Sincerely,
Gizem Issın, MD
(on behalf of all authors)
Reviewer 2 Report
Comments and Suggestions for Authors
This is a timely and useful contribution: appendiceal collision tumors are rare, and combining an institutional series with a systematic literature synthesis is the right approach. The manuscript is generally well-structured and the conclusions are plausible, but several methodological inconsistencies, reporting gaps- please correct the research date, in the abstract may, methods -march 2025. Table 2 “Tumor Stage” portion is confusing (LAMN pTis appears in a header-like position; pT categories are not clearly allocated). Also “Pt4” is inconsistently capitalized.- please could you improve table no. 2. Define all abbreviations at first mention in the main text (even if you have an abbreviation list).
Author Response
Dear Reviewer,
We thank you for your thoughtful and constructive review of our manuscript. We appreciate your recognition of the relevance of the topic and of the combined institutional and systematic review approach, which is particularly important for rare entities such as appendiceal collision tumors.
We have carefully addressed all points raised in your comments. The revisions made are summarized below:
- Correction of research dates
We identified and corrected the inconsistency in the reported study period.
The research end date has now been consistently updated toMay 2025 throughout the manuscript, including the Abstract and the Materials and Methods section. - Revision of Table 2 (Tumor Stage)
We agree that the original presentation of the “Tumor Stage” section in Table 2 was potentially confusing.
Accordingly, Table 2 has been revised to improve clarity and readability:- pT categories are now clearly allocated to their corresponding tumor components.
- The placement ofLAMN pTis has been corrected so that it no longer appears in a header-like position.
- Capitalization inconsistencies (e.g., “Pt4”) have been corrected and standardized topT4.
Overall, the table has been simplified and reformatted to provide a clearer and more consistent presentation of staging data.
- Definition and standardization of abbreviations
All abbreviations have been carefully reviewed and are now defined at their first mention in the main text, in accordance with journal style requirements, even when also included in the abbreviation list.
These revisions have improved the methodological consistency and reporting clarity of the manuscript. We thank you for highlighting these important points, which have strengthened the overall quality and presentation of our work.
Sincerely,
Gizem Issın, MD
(on behalf of all authors)
Reviewer 3 Report
Comments and Suggestions for Authors
Dear Authors,
Thank you for your effort to contribute to the literature. It's nice to see and involve in peer-review of study which conducted by pathologist from multicenter in Türkiye.
Your manuscript presents a valuable institutional case series combined with a comprehensive systematic review of appendiceal collision tumors (ACTs). Given the rarity of ACTs, the integration of original cases with an extensive literature synthesis is a clear strength. The study is generally well structured, methodologically sound, and clinically relevant. The conclusions are appropriately cautious and supported by the data.
The study design, inclusion criteria, and pathological reassessment are clearly described and appropriate.The rationale for focusing on component- and stage-based management is convincing and well supported. No major methodological or interpretive concerns were identified. Only minor corrections requering:
Line ~318 Delete the stray “U”.
Line ~330 “Th Collectively, ACT cases…” Replace “Th” with “Collectively,” or “Thus,”
line ~56m“n contrast to composed neoplasms…” Missing capital “I”
line ~102–103 “was reclassified as goblet cell AC line with current WHO terminology.”
Grammatically awkward. Suggested:
“were reclassified as goblet cell adenocarcinoma in accordance with current WHO terminology.”
Author Response
Dear Reviewer,
We would like to thank you for your careful review of our manuscript and for your positive and constructive comments. We are grateful for your appreciation of the study design, methodological rigor, and clinical relevance of our work, as well as for recognizing the value of integrating an institutional case series with a comprehensive systematic review in the context of this rare tumor entity.
We have carefully addressed all of the minor points you raised, and the manuscript has been revised accordingly. The specific changes are detailed below:
- Line ~318
The stray letter “U” has been removed. - Line ~330
The phrase “Th Collectively, ACT cases…” has been corrected.
It now reads:“Collectively, ACT cases…” - Line ~56
The missing capital letter has been corrected.
“n contrast to composed neoplasms…” has been revised to“In contrast to composed neoplasms…” - Lines ~102–103
The grammatically awkward sentence has been revised in line with your suggestion.
The sentence now reads:“were reclassified as goblet cell adenocarcinoma in accordance with current WHO terminology.”
All corrections were implemented in the revised manuscript, and the text has been carefully rechecked for consistency and clarity.
Once again, we sincerely thank you for your thoughtful review and helpful suggestions, which have contributed to improving the quality and readability of our manuscript.
Sincerely,
Gizem Issın, MD
(on behalf of all authors)